# Spatiotemporal Analysis and Control of Landscape Eco-Security at the Urban Fringe in Shrinking Resource Cities: A Case Study in Daqing, China

**DOI:** 10.3390/ijerph16234640

**Published:** 2019-11-21

**Authors:** Xi Chen, Dawei Xu, Safa Fadelelseed, Lianying Li

**Affiliations:** 1College of Landscape Architecture, Northeast Forestry University, Hexing Road 26, Harbin 150000, China; chenxi90316@nefu.edu.cn (X.C.); choubaoxi@126.com (S.F.); lilianying1982@163.com (L.L.); 2Key Lab for Garden Plant Germplasm Development & Landscape Eco-Restoration in Cold Regions of Heilongjiang Province, Harbin 150000, China; 3Ministry of Agriculture, Horticulture Administration Sector, Landscape and Ornamental Plants Department, Elmogran 999129, Sudan; 4College of Art and Design, Harbin University, Harbin 150000, China

**Keywords:** landscape ecological security, spatial control, urban fringe, resource city, spatiotemporal analysis

## Abstract

As the main bearing area of the ecological crisis in resource-rich cities, it is essential for the urban fringe to enhance regional ecological security during a city’s transformation. This paper takes Daqing City, the largest oilfield in China’s cold land, as an example. Based on remote sensing image data from 1980 to 2017, we use the DPSIR (Driving forces, Pressure, State, Impact, Response) framework and spatial auto-correlation analysis methods to assess and analyze the landscape eco-security change of the study area. From the perspective of time–space, the study area is partitioned, and control strategies are proposed. The results demonstrate that: (1) The landscape eco-security changes are mainly affected by oilfield exploitation and ecological protection policies; the index declined in 1980–2000 and increased in 2000–2017. (2) The landscape eco-security index has obvious spatial clustering characteristics, and the oil field is the main area of warning. (3) The study area determined the protection area of 1692.07 km^2^, the risk restoration area of 979.64 km^2^, and proposed partition control strategies. The results are expected to provide new decision-making ideas in order to develop land use management and ecological plans for the management of Daqing and other resource shrinking cities.

## 1. Introduction

Resource cities, especially resource depleted or reduced cities (such as oilfields), are facing enormous challenges to their ecological environment including: abundance industries and oilfields, densification and abandonment, and pollution and built environment legacies [1,2,3,4]. As an important reserved land resource and an ecological crisis bearing area, the area on the urban fringe (especially in resource-rich cities) is in a period of continuous transition, with diversified land use patterns and a fragile ecological environment, which seriously threatens regional ecological security [5,6,7,8]. Therefore, there is an urgent need to stabilize and enhance the ecological security of the urban fringe in shrinking resource cities.

Landscape eco-security assessment based on land use is the basis for ecological security pattern construction [9,10,11,12]. Although the establishment of a landscape eco-security assessment system is relatively mature, it is seldom used in the urban fringe, especially in resource cities. Based on geographic information systems/remote sensing (GIS/RS) technology, using the pressure–state–response (P-S-R) framework to establish an indicator for the assessment of ecological security, and combining popular assessment methods including hierarchical clustering methods, the comprehensive index method, the analytic hierarchy process (AHP) has been recognized and widely applied worldwide [13,14,15,16]. Since its founding, the PSR framework has also experienced theoretical and methodological development; the DPSIR (Driving forces, Pressure, State, Impact, Response) framework is the outcome of this development [17,18,19]. DSPIR is more suitable for the complex situations of resource cities in this study. In recent years, studies on landscape eco-security assessment system are no longer limited to fixed time points. Moreover, the evolution and trends of landscape eco-security have gradually received attention. An analysis of the causes of changes can provide a strong basis for future policy-making [20,21,22].

The spatial warning and control of landscape eco-security are a direct and effective way to enhance regional ecological security. In recent years, space partitions are becoming more and more detailed, but they are still based on the current situation [23,24], which ignores the dynamics and stability of landscape eco-security. Therefore, in this study, space partitions should consider the spatial and temporal changes of landscape eco-security. The spatial partitioning method is the key to space control. There are two main spatial partitioning methods: One is an overlay method that emphasizes the vertical process between the natural environment, human activities, and land use changes in the landscape unit [25]. The other is a method of using model partitioning to emphasize the landscape level process represented by the minimum cumulative resistance model (MCR). Both methods have a different emphasis and should be given comprehensive consideration in this study [26,27].

At present, there are 262 resource cities in China, while 88% of cities are experiencing or are about to experience recession and transition [28]. This paper takes the urban fringe of Daqing City in Heilongjiang Province as an example. Daqing is a famous petroleum city; however, the peak period of oil production in this city has passed, and the city is about to enter a new period of transformation and development. Therefore, we use Daqing as a forward-looking and representative example for enhancing the landscape eco-security of the urban fringe in a resource city. In order to enhance the landscape eco-security of the urban fringe in Daqing, based on the 1980–2017 landscape eco-security assessment, this paper analyzes the trends of and reasons behind landscape eco-security changes, identifying the spatial control scope and proposing spatial control strategies. This paper’s significance is as follows: (1) We use the DPSIR framework to construct an assessment model and analyze its temporal and spatial variation characteristics, making up for the lack of basic research on the urban fringe of resource cities. (2) Based on time and space analyses, we combine the MCR model and the overlay methods for space partitions, which provides a new partition idea for the space control of resource cities. (3) We analyze the influencing factors of landscape eco-security changes and a proposal for space control strategies to provide support for ecological security construction in the urban fringe of Daqing.

## 2. Materials and Methods

### 2.1. Study Area

The city of Daqing, Heilongjiang Province (123°45′ E–125°47′ E, 45°23′ N–47°29′ N) is located in the hinterland of Songnen Plain, with flat terrain and a relatively concentrated distribution of farmland, grassland, and woodland. Daqing Oilfield, once the largest oilfield in China, is also a world famous super large continental oilfield. Its unique oil resources led to the city’s rise from a pasture with a population of less than 20,000 to a petrochemical city with a population of millions. The study area is located inside the main urban area of Daqing City, outside the central city, and is jointly managed by the oilfield and the administration, with a total area of 2695 km^2^ (Figure 1). According to the administrative division, this area is managed by the Ranghulu District, Saertu District, Longfeng District, and Honggang District. The oilfield mainly includes the second, third, fourth, fifth, and sixth Oil Production Plants of Daqing. The oilfield area is large and distributed in a strip shape. Except for the necessary recourse exploitation demand, the oilfield is hardly affected by other construction land. At present, most of the oil wells are still in service. However, with the exploitation of resources and the development of cities, a large amount of abandoned land will be produced in the study area.

Daqing was established due to the exploration and development of oil fields, so the development of the study area is closely related to the exploitation of oil resources. The area’s oil-derived properties make it unique in its natural resources and land space compared to other cold oil fields in China. With the development of oil and the growth of the local economy, the city’s development has obvious phases of development, which can be divided into three stages: budding, development, and decline and transformation (Figure 2). Before 1960, Daqing was a pasture. After 1960, because of the discovery of oil, the city began to develop. Since 1979, the city officially started construction. After 2000, due to the reduction of oil production and the impact of national environmental protection policies, Daqing has started to formulate a series of ecological environmental protection policies. In the approval of construction projects, laws and regulations, such as the environmental protection law and the Environmental Impact Assessment Law, have been strictly observed. In terms of pollution control, these laws control the total amount and increment of pollution discharge. Emergency plans for heavy pollution weather in Daqing City, as well as an implementation plan for strengthening water and soil pollution control in Daqing City, have been issued. In terms of natural protection, the project of returning grazing land to grassland, rotational grazing, the “Weathered, Sandy and Salinized” land renovation, and afforestation have been carried out. Wetland protection has also been promoted, nature reserve management and protection have been strengthened, several reservoirs in Daqing have been expanded and protected, and the project of controlling a hundred ponds have been implemented since 2000. The ecological spatial pattern of “landscape, forest, farmland, and water” is thus becoming clearer. In oilfield construction, “green environmental protection” has been consistent throughout the process of oil field operations and has gradually attached importance to developing the ecological environment of the oil field. In 2016, a 40 km^2^ oil field ecological construction demonstration area was built.

### 2.2. Data and Land Use Classification Criteria

The remote sensing image data in this study were taken from the United States Geological Survey (USGS). According to Figure 1, the MSS, TM, ETM+, and OLI-TIRS images taken by Landsat satellite in 1980, 1990, 2000, 2010, and 2017 were selected (Table 1). Other auxiliary materials were from Daqing General Planning, the Daqing Statistical Yearbook, the Daqing Oilfield Statistical Yearbook, the China Statistical Yearbook, and the China Agricultural Product Price Survey Yearbook (Table 2).

The land use types in the study area are divided into eight types (Table 3): woodland, high coverage grassland, medium coverage grassland, low coverage grassland, water area, farmland, construction land, saline-alkali land and others (Figure A1). This classification system is based on multi-source land use data, referring to the “Classification of Land Use Status of the People’s Republic of China” (GB/21010–2007), the “China Resources and Environment Database” land use remote sensing classification system of the Chinese Academy of Sciences, and combining remote sensing images with the regional characteristics of the study area.

### 2.3. Methods

#### 2.3.1. A Landscape Eco-Security Assessment System Based on the DPSIR Framework

(1) Establishment of the assessment system and the determination of weight

In this study, the DPSIR framework was used to assess landscape eco-security. In the DPSIR assessment model, “Driving forces” refers to the fundamental driving force and the potential incentive for environmental change, mainly due to the changes brought about by economic and urban development. “Pressure” is the direct cause of the changes in the ecological environment of the research area, which mainly refers to the impact of human activities on the natural environment, such as the exploitation of oil resources and the discharge of waste. “State” is the actual situation of ecological environment under pressure, which is mainly reflected by the proportion and change of grassland coverage and non-ecological land. “Impact” refers to the impact of the state on the ecological environment, which is specifically reflected by the ecological risk, ecosystem service value, and ecological resilience. “Response” refers to policies formulated by human beings to promote sustainable development, such as increasing investment to improve resource utilization efficiency and reducing pollution (Figure 3). According to the specific characteristics and study purpose of the research area, combined with a study of the relevant literature, the assessment system is designed using five dimensions (driving forces, pressure, state, impact, and response), and seven indicators. On the basis of the analytic hierarchy process (AHP) and the suggestions of 15 experts (including those from the fields of urban planning, landscape architecture, ecology, environmental science, and petroleum engineering), the weight of the index layer and the value of the index were obtained (Table 4). In addition, the units and ranges of the selected indicators in the P-S-R framework were different, and the linear normalization function was used to standardize each indicator. The equation is as follows:$$y=g(x)=\frac{x-\min}{\max-\min}$$
where y is a range from 0 to 1, and x is the normalized value;

(2) Index computing method

Combining expert opinions with the relevant references [29,30,31,32,33], the specific index calculation methods are shown in Table 5.

(3) The assessment unit determination

Because the research area is under the jurisdiction of different administrative regions and oilfields, it needs clear analytical units to reflect the specific spatial differences at an objective level. In this paper, the grid and quadrant are two levels of evaluation units (Figure 4). Because there is no uniform standard for the selection of the assessment grid unit [34,35,36], this paper selects a 1 × 1 km grid as the basic unit based on the test of the satellite, the land use classification, and the study area (especially the density of well in oilfield). Taking the center of the study area as the starting point, the area is divided into eight quadrants: north, north-east, east, south-east, south, south-west, west, and north-west.

#### 2.3.2. Spatial Auto-Correlation Analysis

(1) Global auto-correlation analysis

The global spatial auto-correlation describes the spatial aggregation characteristics of the attribute values in the whole study area. Global Moran’s I is the most commonly used analysis index at present [37]. It can measure the aggregation or dispersion degree of the spatial distribution of the landscape eco-security index in the study area. The equation is as follows:$$\mathrm{Moran}’sI=\frac{\sum_{i=1}^{n}\sum_{j=1}^{m}W_{\mathrm{ij}}\left(x_{i}-x\right)\left(x_{j}-x\right)}{S^{2}\sum_{i=1}^{n}\sum_{j=1}^{m}W_{\mathrm{ij}}},\;S^{2}=\frac{1}{n}\sum_{i=1}^{n}{\left(x_{i}-\bar{x}\right)}^{2},\bar{x}=\frac{1}{n}\sum_{i=1}^{n}x_{i}$$
where X_i_ is the observation value of the i-th region, n is the number of the raster, and W_ij_ is a binary adjacent-space weight matrix, indicating the adjacent relationship of the spatial objects. The value range of Moran’s I index is between −1 and 1. Under the premise of a Z-score significance test, if Moran’s I > 0, the attribute values are positively correlated in space, and the larger the index, the denser the spatial distribution. If Moran’s I < 0, the attribute values are negatively correlated in space, and the smaller the index, the more discrete the spatial distribution. If Moran’s I = 0, the attribute values are randomly distributed in space.

(2) Local Spatial Auto-Correlation Analysis

A global spatial auto-correlation analysis can reveal the overall spatial dependence, but for the spatial auto-correlation analysis of larger regions, it is insufficient to analyze the whole area. Therefore, in order to study the possible local spatial aggregation relationship, the hot-spot analysis method and hot-spot optimization analysis (local spatial auto-correlation analysis method) are introduced to explore the spatial aggregation degree of the landscape eco-security index distribution of individual evaluation units.

Hot-spot analysis calculates the Z scores between patches based on the Getis–Ord Gi* statistical index in the GIS platform, which can directly reflect the agglomeration of a high value area (hot-spot area) and a low value area (cold point area) in space [38]. The higher the Z value, the more obvious the agglomeration of the hot-spot area. The equation is as follows:$${\mathrm{Gi}}^{*}=\frac{\sum_{j=1}^{n}w_{\mathrm{ij}}x_{j}-\bar{X}\sum_{j=1}^{n}w_{\mathrm{ij}}}{S\sqrt{\frac{{\left[n\sum_{j=1}^{n}{w_{\mathrm{ij}}}^{2}-\left(\sum_{j=1}^{n}w_{\mathrm{ij}}\right)\right]}^{2}}{n=1}}}$$
where X_j_ is the attribute value of the plaque j; w_ij_ is the spatial weight matrix between the plaque i and the plaque j; and n is the total plaque number.

Hot spot optimization analysis is based on the Optimized Hot Spot Analysis tool (Arcgis 10.5). This tool can interrogate data to automatically select parameter settings that will optimize the hot-spot results. It will aggregate incident data, select an appropriate scale of analysis, and adjust the results for multiple testing and spatial dependence. In this paper, this tool is used to predict the trend of the spatial aggregation of landscape eco-security and identify risk warning areas.

#### 2.3.3. Space Partition Methods

(1) Overlay analysis

The space of the study area was partitioned from the perspective of time–space, considering the past, present, and future landscape eco-security changes of the study area for comprehensive identification and partitioning. The 2010 and 2017 hotspot analysis and 2017 hotspot optimization analysis results were overlaid to identify the ideal protection area, the core protection area, the bottom line protection area, the ecological potential area, the risk supervisory area, the risk prevention area, the key restoration area, and the core restoration area (Figure 5).

(2) Resistance analysis

The resistance surface reflects the trend of the spatial expansion of the ecological flow [39]. This paper uses the minimum cumulative resistance (MCR) model to determine the extent of the protected area. The core protection area, the bottom line area, and the ecological potential area were selected as the ecological source. Based on the hotspot analysis superposition results and the classification of land status, the resistance index system was constructed, whose weight was determined based on 15 experts’ suggestions (Table 6). We calculated the resistance surface using the MCR model and considered the ideal protection area to identity the scope of the protection area. In the protection area, the construction land and farmland were determined as ecological conflict areas, and the spaces that do not overlap with the ecological source were determined as the protection buffer area. The MCR equation is as follows:$$\mathrm{MCR}=f\min\sum_{j=n}^{i=m}D_{ij}\times R_{i}$$
where f is the unknown negative function, indicating the negative correlation between the minimum cumulative resistance and ecological suitability; min represents the minimum value of the cumulative resistance of a landscape unit to the source; D_ij_ is the spatial distance from source j to landscape unit i; and R_i_ is the resistance coefficient of the landscape unit i to the motion process. D_ij_ × R_i_ is used to measure the relative accessibility of a path from its source to a point in space.

## 3. Results

### 3.1. Trend of the Landscape Eco-Security Index

The landscape eco-security index of the study area showed a fluctuating downward trend, in which the index declined in 1980–2000 and increased in 2000–2017 (Figure 6). Among these dates, 1980–1990 is the maximum range of decline, and 2000–2010 is the maximum range of increase. The trend of the impact index is consistent with the trend of the landscape eco-security index. The impact index changes of the different quadrants in the study area show that (Figure 7) the main change period within the oilfields dominated by the north–south direction is 1980–1990. During this period, the index of the north mainly increased due to an increase in the ESV, the index of south mainly declined due to the increase in the index of ERI, and the index of ECO declined. The main change areas on the outside of the oilfields are north-west, west, and south-west. The main change years are 1980–1990 and 2000–2010. Due to the increase in the index of ERI, the landscape eco-security index of the study area showed a downward trend in 1980–1990. Influenced by the decline of the ERI index in the south-west and north-west and the rise of the ESV index in the west, the landscape eco-security index showed an upward trend in 2000–2010.

### 3.2. Spatial Auto-Correlation Trend

#### 3.2.1. Global Auto-Correlation Analysis

From 1980 to 2017, the Moran’s I index > 0 and *p*-value < 0.01. This indicates that the landscape eco-security distribution of the Daqing urban fringe has a significant spatial positive correlation (Table 7). Therefore, the landscape eco-security index in the study area has obvious spatial clustering characteristics.

#### 3.2.2. Local Spatial Autocorrelation Analysis

Hotspot analysis shows (Figure 8), using the method of the spatial fracture point, the spatial aggregation degree (Giz-score) of the study area is divided into seven grades: cold spots-high, cold spots-middle, cold spots-low, not significant, hot spots-high, hot spots-middle, and hot spots-low. The proportion of the hot-spot area was larger than that of the cold spot in 1980–2017 (Figure 9). The hotspot areas decreased year by year, and the cold spot areas continued to rise. This indicates that although the landscape eco-security increased with fluctuations, the low index aggregation areas continue to increase. The main changes occurred in 1980–1990 and 2000–2010. Therefore, we should control the area of low aggregation’s degree in the future to prevent an increase of its area.

The hot-spot optimization analysis shows that there is a large coupling between the cold-spot gathering location and the oilfield location (Figure 10), which is the future risk warning area.

According to the quadrant analysis (Figure 11), the cold-spot proportion is mainly concentrated in the south, south-east, north, and north-east. Among these areas, cold spots-high are mainly distributed in north and south. The hotspots distribution is opposite that of the cold spots. Hotspots are mainly concentrated in the west, south-west, north-west, and east. Among these areas, the hot spots-high proportion is high in the west and northwest. The change of aggregation is mainly concentrated in the south, south-west, west, east, north-east, and south-east. North of the cold-spot proportion continues to increase, while the hot spot continues to decrease where the oilfield is located. West of the hot-spot proportion increased, while the cold-spot proportion decreased in 1980–1990. West of the hot-spot proportion decreased in 2000–2010. The south-west trend is opposite to that of the west. Both the area north-east of the cold-spot proportion and the hot-spot degree increased in 1980–1990; the 2000–2010 change of the north east is opposite to that of 1980–1990. The area south-east of the cold-spot proportion decreased, while the hot spot increased in 1980–1990. The area south-east of the cold-spot proportion increased, while the hot-spot proportion decreased in 1990–2017. The area east of the cold-spot proportion increased, while the hot-spot proportion decreased in 1980–1990.

### 3.3. Space Partition

Based on the 2010–2017 hot-spot analysis and the hot-spot optimization analysis in 2017, we identified a risk warning area of 979.64 km^2^, of which the core restoration area is 46.54 km^2^, the key restoration area is 241.98 km^2^, the risk prevention area is 106.39 km^2^, the risk supervision area is 357.13 km^2^, and the ecological potential area is 227.59 km^2^. We identified the bottom line protection area as 106.29 km^2^, while the core protection area is 222.85 km^2^, and the ideal protection area is 296.76 km^2^ (Figure 12). The bottom line protection area and the core protection area are identified as ecological sources, and the ecological potential area is identified as a potential ecological source. According to the resistance analysis result and the ideal protection scope, the areas around the core source resistance of <30,000 and the potential source resistance of <14,000 identified the protection area (1692.07 km^2^), of which the ecological buffer area is 1135.34 km^2^, and the ecological conflict area is 496.52 km^2^ (Figure 13).

According to an analysis of land use type (Figure 14), the main land use type in the core restoration area is saline-alkali land, which is identified as the main risk source. The key restoration area is mainly composed of saline-alkali land and farmland, which is concentrated at the edge of the core restoration area in the south of the oilfield. The risk prevention area is mainly composed of saline-alkali land, farmland, and medium-coverage grassland. The farmland is concentrated at the edge of the saline-alkali land in the southern area of the oilfield, and its risk is relatively high. The medium coverage grassland is located at the edge of the saline-alkali land in the north of the oilfield. The grassland is easily degraded and converted into saline-alkali land. The saline-alkali land is concentrated at the edge of the key restoration area, whose scope is small and interlaced with other land. The risk supervision area is mainly composed of farmland, which is located around the risk prevention area and plays a buffering role. The ecological potential area is mainly composed of farmland and grassland and is located in the south of the oilfield. The bottom line protection area is mainly composed of water in the north-east of the study area. The core protection area is dispersed outside the oilfield area. The main land use type is high-coverage grassland. The ecological potential area is located inside the oilfield, and its main land use types are medium-coverage grassland and farmland. The ideal protection area and the protection buffer connect the core protection area in series. The main land use types are farmland and high coverage grassland.

## 4. Discussion

### 4.1. The Main Reasons for Landscape Eco-Security Change

According to the curve fittings (Figure 15) [40,41,42], the changes of the landscape eco-security index in the study area are affected by the socio-economic indicators (GDP, ecological investment, oil production), the landscape patches number, and the land use types (saline-alkali land, high coverage grassland, and water).

At present, we can only collect the overall construction and environmental data for Daqing City, while the specific data for the Daqing urban fringe area and each plant area are deficient, which makes it impossible to carry out a detailed analysis. Therefore, the analysis of the reasons for the landscape eco-security changes in each quadrant is mainly based on the impact layer index, the hot-spot analysis and land use transfer (Figure 16). These changes reflect four phenomena: (1) A lack of protection control of the study area has caused the landscape’s eco-security to be destroyed. Due to the lack of farmland control, disorderly reclamation destroyed a large number of high-coverage grassland areas and increased the fragmentation of green land, thereby decreasing the 1980–1990 impact indices of the west and south-west, while the north-west increased the 1980–1990 cold-spot proportions of the south-west, east, and north-east and the 2000–2010 cold-spot proportion of the west. Due to the lack of construction land control, the expansion of construction land destroyed the original green space, and the cold-spot proportion of the south and south-east continued to increase after 1990. (2) Oilfield exploitation is mainly manifested through two trends. On the one hand, oilfield exploitation causes surface subsidence. Wetlands and potholes were formed in and around the oilfield, which increased the impact indices of the north and the hot-spot proportion of the north-east in 1980–1990. On the other hand, the exploration and expansion of the oilfield destroyed the surface vegetation, which eventually degraded into saline-alkali land, and the coverage of the grassland around the saline-alkali land continued to decrease due to a lack of protection and renovation. Therefore, the exploitation of oilfields has formed a large-scale risk warning area, which decreased the impact indices of the south in 1980–1990 and increased the cold-spot proportion of the south in 1980–2017. (3) Environmental protection policies, such as rotational grazing, returning farmland to green land, and water restoration, have improved the grassland coverage, the blue–green space area, and coherence, which greatly improved the landscape eco-security of the external oilfield. The impact indices of the south-west, west, and north-west, as well as the hot-spot proportion of the south-west and north-east, increased in 2010–2017. Therefore, in the future construction of the urban fringe of Daqing, to implement existing environmental protection policies, it will be necessary to strictly control the risk warning area and the protection area.

Therefore, in the future, we should reduce the risk warning area and the landscape patches number; increase the ecological protection areas; and develop ecological industries to increase GDP and ecological investment.

### 4.2. Risk Warning Area Control Strategies

The control strategy for the risk warning area should focus on the restoration and prevention of the risk source (saline-alkali land, which is mainly formed by the oilfield exploitation without ecological control). For the core restoration area and the key restoration area, since the saline-alkali land has been left for many years, its degree of salinization is serious, so it is difficult and time-consuming to reuse it. In future development, this land should be mainly used for construction. If necessary, after detecting the salinity of the area, a traditional treatment method (physical and chemical treatment) is used for the classification treatment [43]. The risk prevention area should focus on controlling the further degradation of land. The control strategy for farmland is as follows [44]: (1) Reduce the use of chemical fertilizers to prevent land salinization and advocate the use of farmyard manure, such as straw returning and livestock manure, to improve soil organic matter content. (2) Under certain conditions, the fallow and returning farmland will promote the recovery rate of soil fertility, and farmers who take turns to fallow farmland resources should be subsidized to promote the recovery of soil fertility. Optimized use after returning the farmland is mainly concentrated on returning the farmland to forests and grasslands. (3) Integrate the farmland to reduce the landscape fragmentation. Integrate the farmland into blocks, gather the farmland together and improve connectivity between the farmland, thereby enhancing its landscape eco-security. The control strategy for medium coverage grassland is as follows [45]. (1) If it is discontinuous grassland with a small area, it should be developed into farmland, which will reduce the fragmentation of the farmland. (2) For large-scale continuous grassland, human damage, such as grazing, should be strictly restricted. As an important area to reduce the regional landscape’s ecological risk and protect the landscape’s ecological environment, the risk supervision area should focus on maintaining the stability of the landscape, such as planting salt tolerant crops or developing improved rice planting, to improve economic benefits while achieving risk supervision [46]. The ecological potential area has both a potential salinization risk and resistance to salinization. The landscape in this area is relatively stable and has high ecological potential. Therefore, it is important to focus on its protection and utilization. Appropriate development should be carried out while exerting the ecological potential of the land, so as to optimize the allocation of land resources. Using the ecological potential area to establish a blue ecological network can reduce the total soil salinity in the study area [47].

### 4.3. Ecological Protection Area Control Strategies

The control strategy for the protection area should focus on restricting the ecological conflict area to protect the ecological source (Table 8). For the ecological protection core area, in principle, any construction projects are prohibited from entering—that is, all ecological resources in the area are protected, including enclosure restoration, and all activities that may damage the ecological environment are prohibited [48]. For the water bottom line protection area, except for the ecological protection core area control strategies, the demarcation and isolation protection of drinking water should be carried out at the same time, and control measures, such as habitat restoration, proliferation, and release of the key habitats of aquatic organisms, should be implemented [49]. The ideal protection area has the potential to develop into a core protection area. Therefore, priority should be given to protecting ecology, strictly controlling the development and construction of projects that destroy ecological functions, and encouraging the establishment of green infrastructure, such as ecological green corridors, country parks, and rural landscapes, thus returning farmland to green and demolishing non-ecological buildings (only retaining single-story ecological buildings as green infrastructure housing) [50]. For the ecological potential area, in addition to the restrictions of the ideal protection area, the potential value of ecosystem services can be appropriately exerted to develop the farmland for ecological farm projects [51]. As a transitional area between ecological space and non-ecological space, the ecological buffer zone is less restricted (retain low-rise buildings). In addition to the construction that can be carried out by ecological potential areas, the ecological agriculture project, and the necessary rural living and supporting service facilities, production infrastructure for cultivation, eco-tourism, and leisure facilities are also encouraged to be built [52].

## 5. Conclusions

Through the application of remote sensing and GIS technologies, and using the DPSIR framework to build an assessment model, this paper analyses the trends of and reasons behind landscape eco-security changes in the urban fringe of Daqing from 1980 to 2017, thereby determining the area’s spatial partition scope and control strategies. The landscape ecological security index shows a downward trend. In order to enhance the landscape eco-security of the study area, two types of control strategies are proposed: controlling the expansion of risk sources (the risk warning area) and protecting the ecological space (the protection area). Conclusions were drawn and are presented as follows.

The landscape eco-security index of the study area was bounded by the year 2000. From 1980 to 2000, the index of the study area decreased (especially the oilfield); from 2000 to 2017, the index of the external oilfield increased. The landscape eco-security index in the study area has obvious spatial clustering characteristics. The proportion of cold spots increased and formed a risk warning area. The changes of the landscape eco-security index in the study area is affected by the socio-economic indicators (GDP, ecological investment, and oil production), the landscape patches number, and the land use types (saline-alkali land, high coverage grassland, and water).

Based on the result of the time and space analysis, we identified a risk warning area of 979.64 km^2^ (mainly in the oilfield), including an ecological potential area, a risk supervisory area, a risk prevention area, a key restoration area, and a core restoration area. The protection area of 1692.07 km^2^, includes the bottom line protection area, core protection area, ideal protection area, ecological buffer area, and conflict area. Risk warning area strategies focus on controlling and renovating saline-alkali land (the risk source) to prevent further expansion. Ecological protection area strategies focus on controlling the disorderly expansion of construction land and farmland, increasing the coherence and area of the blue–green space, and improving grassland coverage.

## Figures and Tables

**Figure 1 ijerph-16-04640-f001:**
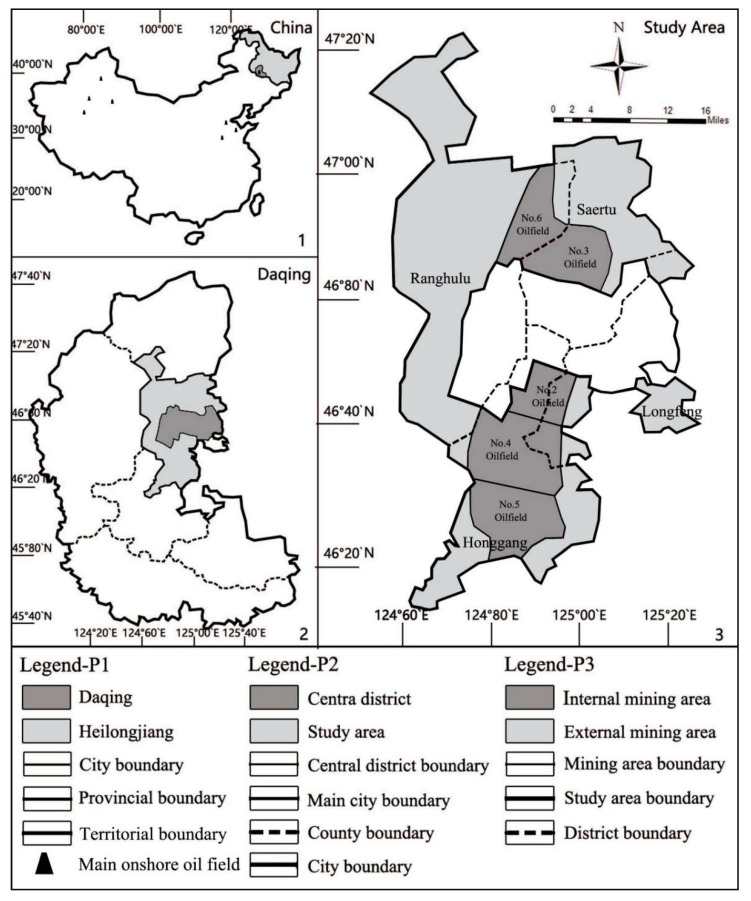
Location of the study area.

**Figure 2 ijerph-16-04640-f002:**
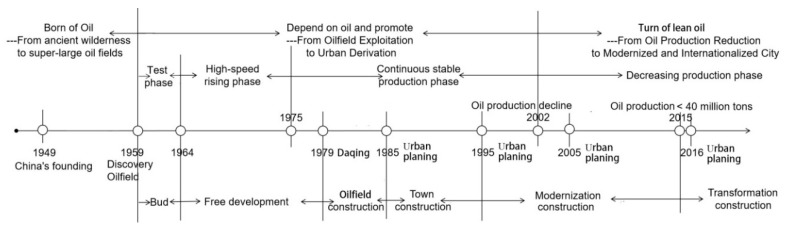
Urban development context of the study area.

**Figure 3 ijerph-16-04640-f003:**
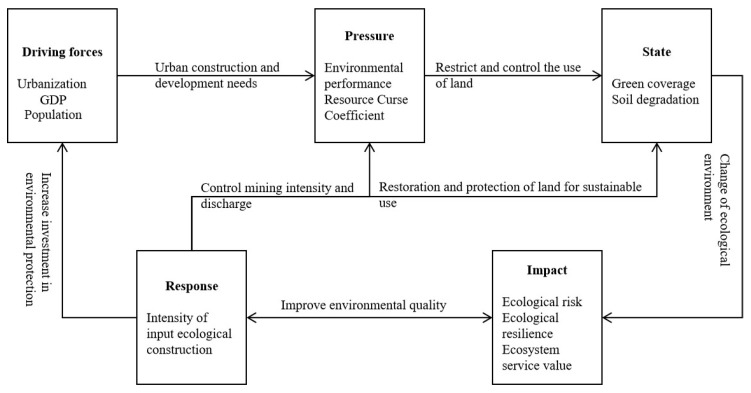
DPSIR model of ecological environment quality evolution in the study area.

**Figure 4 ijerph-16-04640-f004:**
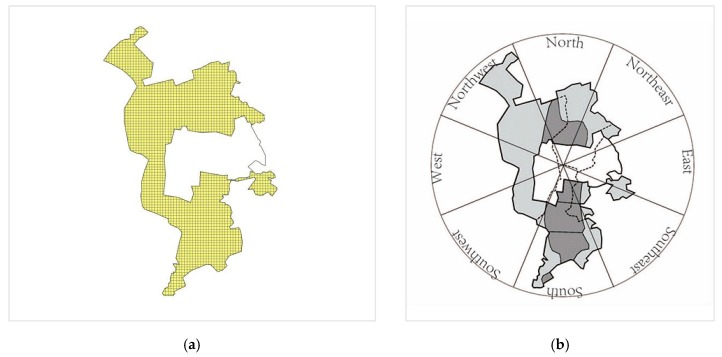
Determination of the area of analysis (**a**); 1 × 1 km assessment unit (**b**); quadrant division of the study area.

**Figure 5 ijerph-16-04640-f005:**
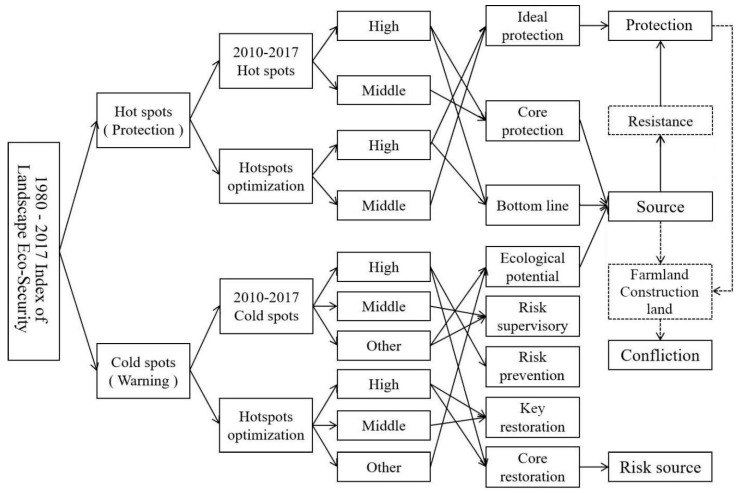
Partitioning standards.

**Figure 6 ijerph-16-04640-f006:**
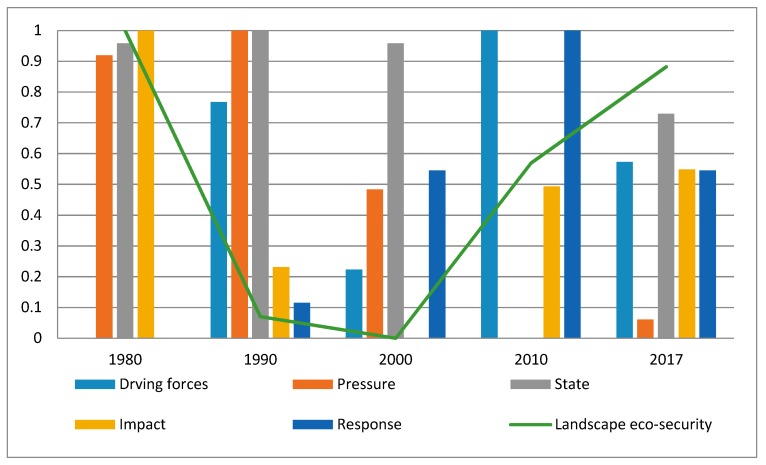
Changing trend of the landscape eco-security index in the study area.

**Figure 7 ijerph-16-04640-f007:**
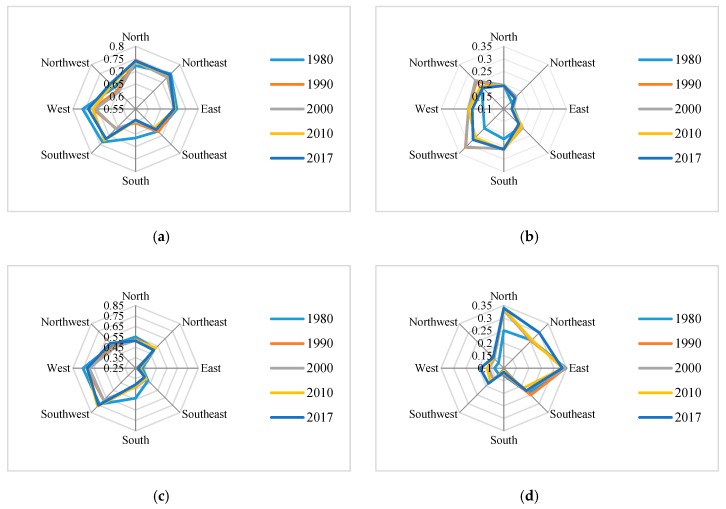
Landscape impact layer index changes in 8 directions (**a**). Trend of the impact index in 8 directions (**b**). Trend of the ERI index in 8 directions (**c**). Trend of the ECO index in different quadrants (**d**). Trend of the ESV index in 8 directions.

**Figure 8 ijerph-16-04640-f008:**
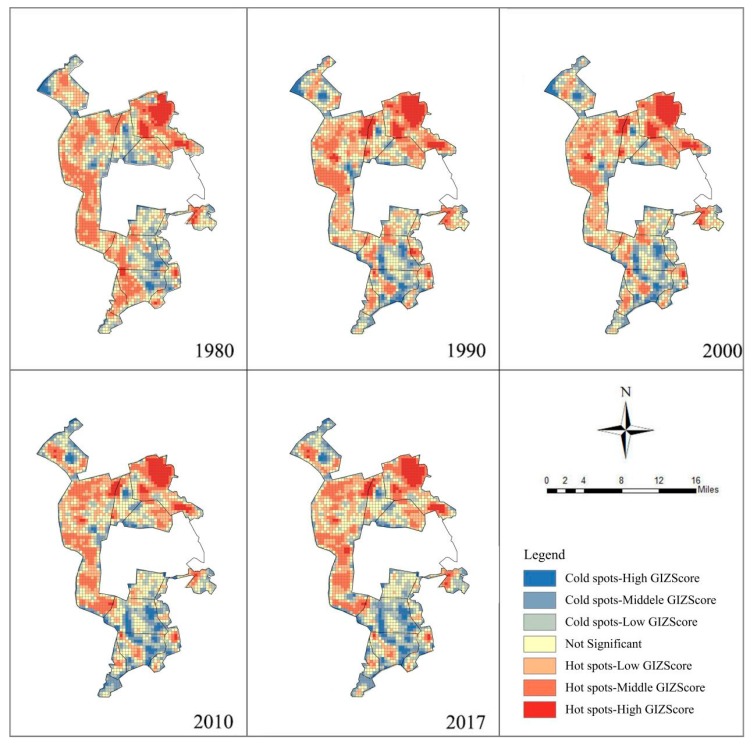
Local spatial autocorrelation analysis in the study area.

**Figure 9 ijerph-16-04640-f009:**
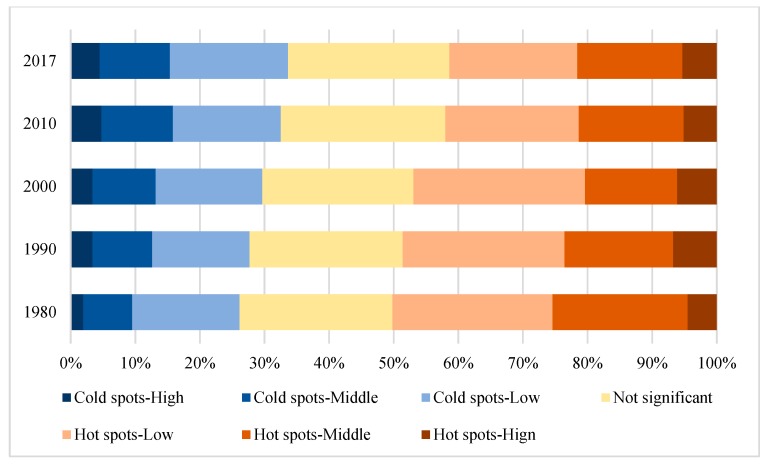
Change in the proportion of spatial agglomeration in the study area.

**Figure 10 ijerph-16-04640-f010:**
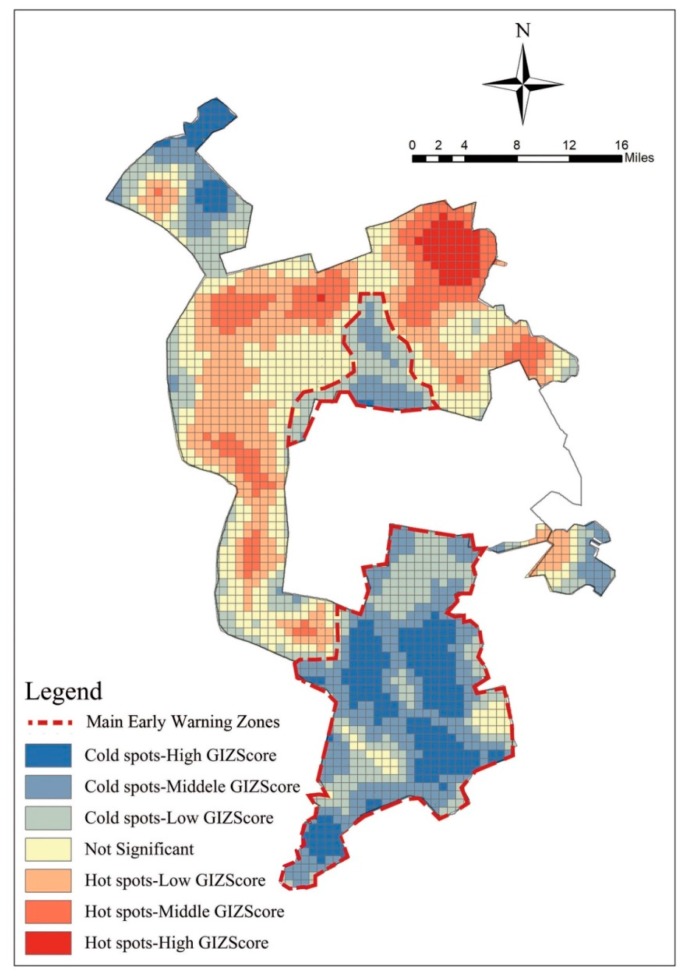
Hot-spot optimization analysis.

**Figure 11 ijerph-16-04640-f011:**
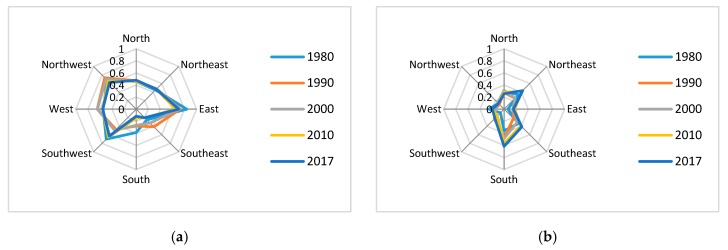
Change of spatial diversity in 8 directions (**a**). Proportion of the hotspot in 8 directions (**b**). Proportion of the cold spots in 8 directions (**c**). Proportion of the hot spot-high area change in 8 directions (**d**). Proportion of the cold spot-high area’s change in 8 directions (**e**). Proportion of the hot spot-middle area’s change in 8 directions (**f**). Proportion of the cold spot-middle area change in 8 directions (**g**). Proportion of the hot spot-low area change in different quadrants (**h**) Proportion of the cold spot-low area change in 8 directions.

**Figure 12 ijerph-16-04640-f012:**
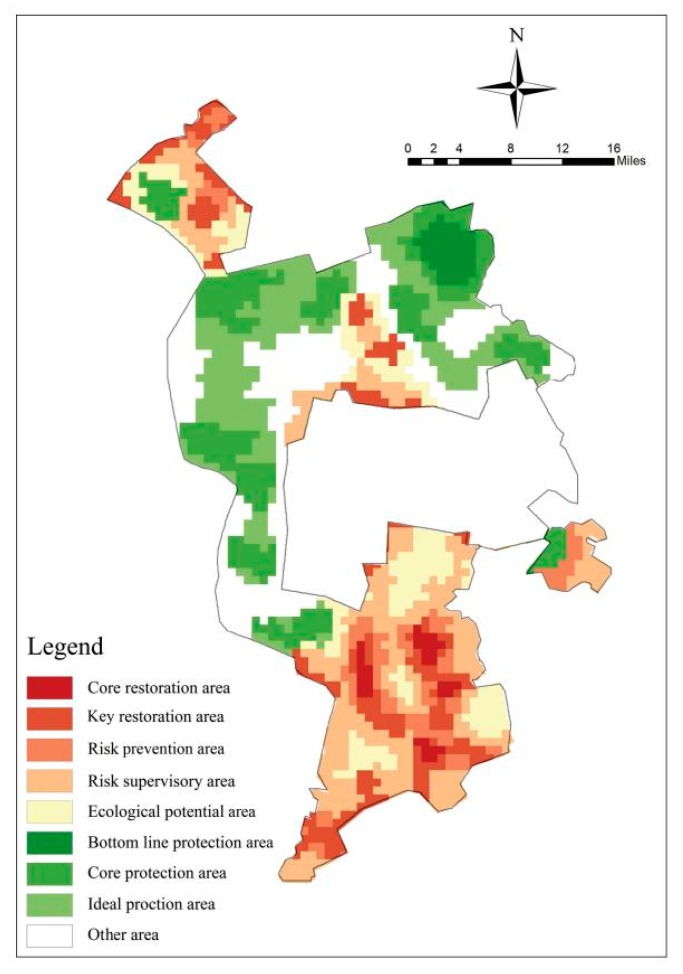
Time–space overlay analysis.

**Figure 13 ijerph-16-04640-f013:**
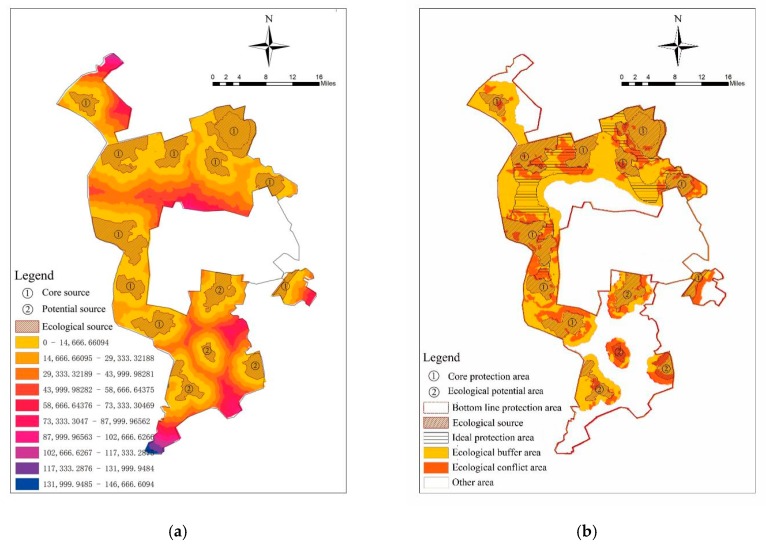
Protection area analysis: (**a**) resistance analysis; (**b**) protection area partition.

**Figure 14 ijerph-16-04640-f014:**
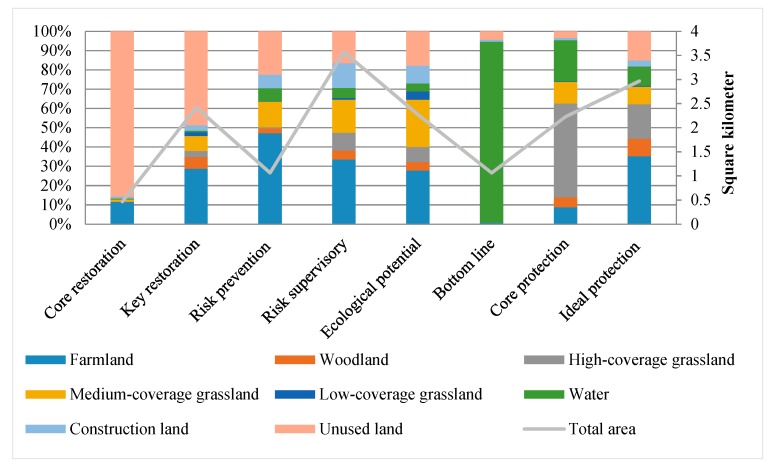
Space partition and land use type.

**Figure 15 ijerph-16-04640-f015:**
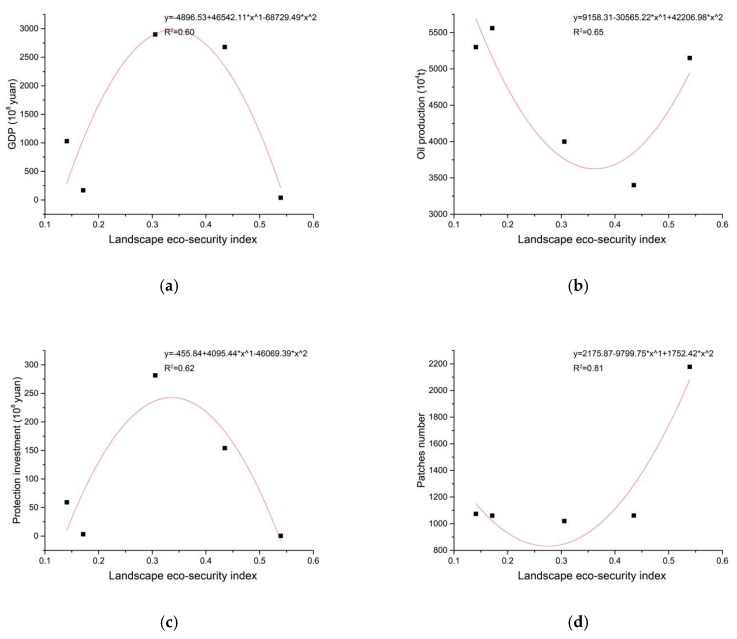
The relationship between landscape eco-security index and the indicators: (**a**) GDP indicators; (**b**) oil production indictors; (**c**) protection investment indicators; (**d**) landscape patches indicators; (**e**) water and high coverage grassland area indicators; (**f**) saline-alkali land area indicators.

**Figure 16 ijerph-16-04640-f016:**
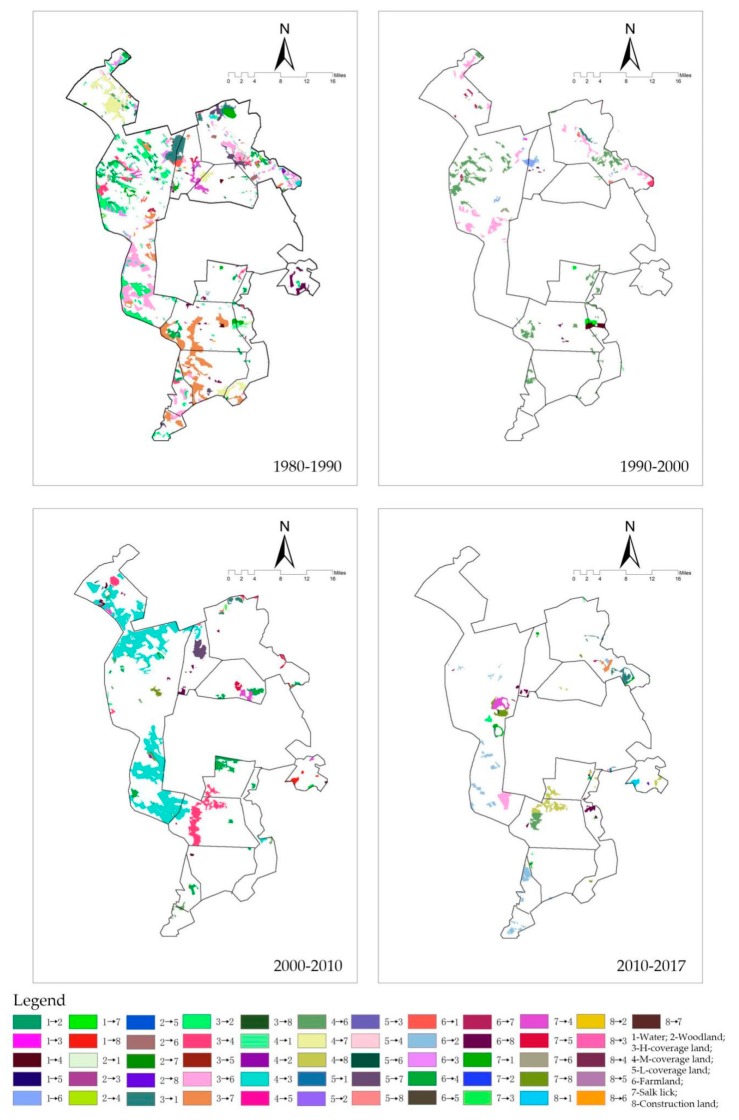
Distribution of land use transfer.

**Table 1 ijerph-16-04640-t001:** Remote sensing information at different times.

Satellite	Date	Data Type	Band	Resolution	Cloud	Kappa
LANDSAT1–3	1980.07	MSS	4	30 M	0	83.3%
LANDSAT4–5	1990.07	TM	7	30 M	0	85.4%
LANDSAT7	2000.07	ETM+	7	30 M	0.01	86.8%
LANDSAT8	2010.07	OLI–TIRS	11	30 M	0	87.1%
LANDSAT8	2017.07	OLI–TIRS	11	30 M	0.01	87.8%

**Table 2 ijerph-16-04640-t002:** Socioeconomic information 1980–2017.

Name	Contents
Daqing Urban Planning	The boundary of the urban fringe; the specific content of the plan (especially in ecological environment construction)
Daqing Statistical Yearbook	Oilfield production; population; GDP; pollution consumption; the primary, secondary, and tertiary output value; environmental protection investment; population;
Daqing Oilfield Statistical Yearbook	Oilfield scope; oilfield ecological construction plan;
China Statistical Yearbook	GDP; mineral production (especially oilfield); pollution consumption; population;
Heilongjiang Agricultural Product Price Survey Yearbook	Planting area of grain crops in Daqing; total output value of grain crops in Daqing;

**Table 3 ijerph-16-04640-t003:** Classification system of land use in the study area.

I	II	Description
Ecological land	Woodland	Forest land; shrub land; open forest land;
	High coverage grassland	Grassland coverage >50%;
	Medium coverage grassland	Grassland coverage ranges from 20% to 50%;
	Low coverage grassland	Grassland coverage ranges from 5% to 20%;
	Water	Reservoirs; ponds; lakes; marshes;
Non-ecological land	Farmland	Dry land; paddy field;
	Construction land	Rural; Urban; industrial and transportation land;
	Saline-alkali land and others	Saline-alkali land; other unused land;

**Table 4 ijerph-16-04640-t004:** Landscape ecological security assessment system of the urban fringe in Daqing.

Dimension	Weight	Sub-Dimension	Indicator	Weight	(±)
D—Driving forces	0.05	Urbanization	D1—Urbanization growth intensity (UGI)	0.5	(+)
		Economy;	D2—Per capita GDP	0.5	(+)
P—Pressure of landscape change	0.2	Oil production	P1—Resource Curse Coefficient (ES_i_)	0.5	(+)
		Environment	P2—Environmental performance index (EPI)	0.5	(+)
S—State of land use	0.2	Ecological land	S1—Grassland degradation intensity (K_i1_)	0.25	(−)
			S2—Grassland restoration intensity (K_i2_)	0.25	(+)
		Non-ecological land	S3—Proportion of non-ecological land (Ui)	0.5	(−)
I—Impact of the landscape ecological system	0.5	Risk	I1—Ecological risk index (ERI_k_)	0.5	(−)
		Resilience	I2—Ecological resilience (ECO_res_)	0.25	(+)
		Service	I3—Ecosystem service value (ESV)	0.25	(+)
R—Human response	0.05	Human	R1—Intensity of input ecological construction (IEC)	1	(+)

**Table 5 ijerph-16-04640-t005:** Index computing method for landscape ecological security assessment.

Indicator	Equation	Description
D1	$\mathrm{UGI}=\frac{U_{\mathrm{bd}}-U_{\mathrm{ad}}}{U_{b}-U_{a}}$	U_bd_ is the urbanization index of the study city d in year-b, U_ad_ is urbanization index of the study city d in year-a, U_b_ is the national urbanization index in year-b, and U_a_ is the national urbanization index in year-a.
P1	$ES_{i}=\frac{E_{i}/\sum_{i=1}^{n}E_{i}}{SI_{i}/\sum_{i=1}^{n}SI_{i}}$	E_i_ is the resource production in region i, SI_i_ is the output value of secondary industry in region i, and n is the number of regions.
P2	$\mathrm{EPI}=\frac{x_{i}/g_{d}}{X_{i}/G}$	x_i_ is the total consumption of urban i or pollutant i of the study city d; X_i_ is the total consumption of urban i or pollutant i in China; g_d_ is the GDP of the study city d; and G is the national GDP.
S1	$K_{i1}=\frac{\Delta S_{it1}}{S_{\mathrm{it}}}\times \frac{1}{\Delta t}$	ΔS_it1_ is the area of grassland transferred to lower coverage grassland, construction land, saline–alkali land, and others. S_it_ is the area of grassland at the start of the study, and Δt is the time interval.
S2	$K_{i2}=\frac{\Delta S_{it2}}{S_{\mathrm{it}}}\times \frac{1}{\Delta t}$	ΔS_it2_ is the area of grassland transferred to higher coverage grassland, water, and woodland. S_it_ is the area of grassland at the start of the study, and Δt is the time interval.
S3	$\mathrm{Ui}=\frac{S_{i}}{A}$	S_i_ is the area of non-construction land. A is the total area of sampling block i.
I1	$ERI_{k}=\sum_{i=1}^{n}\frac{A_{\mathrm{ki}}}{A_{k}}\left(E_{i}\times F_{i}\right)$	A_ki_ is the area of land use type i in study area k, A_k_ is the area of study area k, E_i_ is the interference index of land use type i (Table A1), F_i_ is the vulnerability index of land use type i (Table A2), and n is the number of land use types.
I2	$ECO_{\mathrm{res}}=\sum_{i=1}^{n}\left(\frac{A_{i}\times P_{i}}{C_{i}}\right)$	A_i_ is the area of land use type i in study area, P_i_ is the elastic score of land use type i, and n is the number of land use types. C_i_ is the landscape fragmentation index of land use type i (Table A3).
I3	$\mathrm{ESV}=\sum_{i=1}^{n}A_{i}\times VC_{i}$	A_i_ is the area of land use type i in study area, VC_i_ is the total value coefficient of ecological function per unit area of land use type i (Table A4), and n is the number of land use types.
R1	$IEC_{d}=EI_{d}/g_{d}$	EI_d_ is the amount of investment in ecological construction of the study city d, and g_d_ is the GDP of the study city d.

**Table 6 ijerph-16-04640-t006:** Establishment of the resistance surface.

Dimension	Weight	Indicator	Resistance Value
Land use	0.5	Water	1
		Woodland	2
		High coverage grassland	3
		Medium coverage grassland	4
		Low coverage grassland	5
		Farmland	6
		Saline–alkali land and others	7
		Construction land	8
Overlying result	0.5	Core protection area (including Bottom line protection area)	1
		Ecological potential area	2
		Ideal protection area	3
		Other area	4
		Risk supervisory area	5
		Risk prevention area	6
		Key restoration area	7
		Core restoration area	8

**Table 7 ijerph-16-04640-t007:** Results of the global auto-correlation analysis in the study area.

Global Auto-Correlation Index	1980	1990	2000	2010	2017
Moran’s Index	0.473113	0.509808	0.493504	0.473886	0.502602
Expected Index	−0.000427	−0.000427	−0.000427	−0.000428	−0.000428
Variance	0.000234	0.000241	0.000234	0.000235	0.000234
Z-score	30.932970	32.874252	32.260210	30.921220	32.884483
*P*-value	0.000000	0.000000	0.000000	0.000000	0.000000

**Table 8 ijerph-16-04640-t008:** Ecological conflict area control strategies.

The Protection Area	Farmland	Construction Land	Development Proposals
Core protection area; Bottom line area	Returning farmland to a green–blue space	Dismantle	No construction, only for scientific research and education
Potential ecological area	Retain and convert to ecological farmland	Retain single-story ecological buildings	green infrastructures, such as ecological greenways, country parks, and country landscapes The ecological agriculture project,
Ideal protection area	Returning farmland to a green–blue space	Retain single-story ecological buildings	Construction of green infrastructures
Ecological buffer	Retain and convert to ecological farmland	Retain low-rise buildings	The ecological agriculture project; necessary rural living service facilities; cultivation production infrastructure; eco-tourism; leisure facilities

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
