# Peer review of "Spatiotemporal Analysis and Control of Landscape Eco-Security at the Urban Fringe in Shrinking Resource Cities: A Case Study in Daqing, China"

_ijerph, 2019, doi:10.3390/ijerph16234640_

Round 1
Reviewer 1 Report
In this manuscript, the author(s) assesses the landscape ecological security of the urban fringe area of a resource city in China by spatial analysis and the Pressure(P)-State(S)-Response(R) framework. It contributes to the literature by applying PSR to the urban fringe region in resource city. I would like to recommend a minor revision and my comments are as follow:
1) Since the founding of the PSR framework, it has also experienced theoretical and methodological development, of which the DPSIR (Driving forces, Pressure, State, Impact, Response) framework is the outcome of such development [1,2]. This study could be improved by replacing the current utilization of PSR framework in Daqing city with the DPSIR framework [3]. Or discuss the applicability of DPSIR and limitations of applying it to the study area in the discussion part.
2) The PSR framework is largely built on expert system, from the selection of indicators to the decision of element weights. The sources of experts’ suggestions in line 134 should be explained in more details, e.g. how many experts and what are their expertise areas.
3) The second half of the results section is not well presented and need to be improved in two points: a) the optimization method for hot spot results and the basis and literature that supports the resistance surface building (Table 5) and partition standard (Fig.24) is not introduced in the methods section and the section 4.3 is poorly organized; b) the figure order and presentation is confusing. Fig 15 falls between Fig. 24 and Fig. 25. The title “Figure 16” should be “Figure 26” based on text. Figure 24 and 25, as well as Figure 27, are not indexed in text and there is no corresponding description or analysis in text.
4) The Moran’s I values in line 221 should be provided with the significance level, are they significant at the 0.01 or 0.05 level?
5) Since the protection policies are one of the important drivers in the evolution of ecological security index, the national and local policies implemented in the past decades should be summarized and introduced in the study area section.
6) Consider to move the study area map to figure 1 and index it in line 77.
7) Unify the reference style. Some used abbreviated given name but some are full given name.
8) Avoid grammar and expression errors. For example, “The results shows…” in the abstract. And the “…stability of the plaque” in line 56 and “…degree of plaque fragmentation” in line 363 seem Chinglish and are not normal expression for fragments/segments.
[1] Smeets, Edith, and Rob Weterings. "Environmental indicators: Typology and overview." (1999).
[2] Gari, Sirak Robele, Alice Newton, and John D. Icely. "A review of the application and evolution of the DPSIR framework with an emphasis on coastal social-ecological systems." Ocean & Coastal Management 103 (2015): 63-77.
[3] Wang, Zhen, et al. "A DPSIR model for ecological security assessment through indicator screening: a case study at Dianchi Lake in China." PloS one 10.6 (2015): e0131732.
Author Response
Dear reviewer,
Thank you for your comments of this article. These comments have benefited us a lot. We revised the comments one by one.Please see the attachment(Including the reply commments and the revised manuscript) .
Kind regards,
Dawei Xu

Reviewer 2 Report
I have read this manuscript with ample interest. Whilst the thematic area is timely, current version of the manuscript has many lacking which must be addressed to improve its international significance. I have provided my comments below that may be of help for the authors.
[1] There is no focus of the work, I had difficult time to understand as to how this work is important to contribute to literature. As such may things are not clear nor it does have specific objective(s) to address
[2] Methods of the work are not well structured, some of them require reworking. The workflow does not align with the objective(s) at hand
[3] Discussion and conclusion sections are not clearly linked with research questions at hand
[4] Moderate English corrections are required
[5] Too many objectives at hand to solve hence lacking clear focus
[6] Fig. 1 does not clearly portray what you want to show here, very unclear illustration
[7] Line 110: should it be only ‘Remote Sensing’?
[8] 1km x 1km: what is the basis of this? If I choose more than this what happens? What is the sensitivity of having differing resolution grids, say 5x5 km or more than that? Nothing is clarified, should clearly stated in the method section
[9] Fig. 6 should be ‘changing trends’
[10] Table 5: suddenly you introduced this here without telling readers anything beforehand, should detail in the method section. What is the rationality of using this table and related texts? Must clarify
[11] Conclusion section does not reflect objective(s) at hand
Author Response

(The authors gave the same response as above.)

Round 2
Reviewer 2 Report
Thanks for taking my comments into account to improve the original submission. Despite the revision has been improved, I still feel that motivation should be improved. As such you may consult the following works to reinforce your motivation and discussion of the results obtained through this work.
https://www.sciencedirect.com/science/article/pii/S0143622813001823
https://www.mdpi.com/2072-4292/11/2/105
https://link.springer.com/article/10.1007/s10708-010-9399-x
Author Response
Dear Reviewer,
Thank you again for your help to the article. We have carefully studied the reference article you provided. In the method part, we reset up the driving forces of the assessment system from the urbanization, population, and GDP; in the discussion part, we add the non-linear fittings and land use transfer to better analyze the reasons for the change of landscape eco-security in the study area. Please see the attachment for specific replys.
Kind regards,
Dawei Xu
